# Non-interferometric stand-alone single-shot holographic camera using reciprocal diffractive imaging

Jeonghun Oh [1,2], Herve Hugonnet[1,2] & YongKeun Park [1,2,3] ✉

An ideal holographic camera measures the amplitude and phase of the light field so that the focus can be numerically adjusted after the acquisition, and depth information about an imaged object can be deduced. The performance of holographic cameras based on reference-assisted holography is significantly limited owing to their vulnerability to vibration and complex optical configurations. Non-interferometric holographic cameras can resolve these issues. However, existing methods require constraints on an object or measurement of multiple-intensity images. In this paper, we present a holographic image sensor that reconstructs the complex amplitude of scattered light from a single-intensity image using reciprocal diffractive imaging. We experimentally demonstrate holographic imaging of three-dimensional diffusive objects and suggest its potential applications by imaging a variety of samples under both static and dynamic conditions.

A conventional image sensor cannot perceive the depth of an object. Therefore, it only shows a projected image of the intensity distribution. As the optical focus cannot be changed, information is lost owing to the blurring of the out-of-focus parts of the image. By contrast, a holographic camera provides volumetric information about a three-dimensional (3D) object by measuring its amplitude and phase. This unique capability of holographic cameras is utilized in various fields, including 3D vision systems[1–6] and microscopic imaging of unlabeled samples[7–11]. Moreover, holographic cameras can be used in conjunction with holographic displays or virtual reality technology to improve realism[12,13].

Widely adopted holographic-field measurement methods such as off-axis[14] or phase-shifting holography[15] employ interferometry to obtain the phase information of an object. However, these methods often require complex optical systems. By contrast, non-interferometric or intensity-based techniques enable holographic measurements without using a reference beam and allow simpler and more stable optical setups[16,17]. Fourier ptychographic microscopy (FPM)[18,19] has emerged as a popular intensity-based holographic method. FPM reconstructs an optical field by measuring the intensity distributions generated from various incident angles. It

iteratively maps the Fourier transform of the acquired images in Fourier space.

Reflective FPM exploits the reflection geometry for imaging diffusive samples[20–22]. Holloway et al.[23]. successfully retrieved the field scattered from optically rough samples using reflective FPM. Through a synthetic-aperture-based method, objects that we see in our daily lives can be holographically recorded in a reference-free regime. However, FPM requires redundant intensity images because the Fourier spectra of the different measurements need to be superimposed to reconstruct an optical field. This algorithm increases the data acquisition time. Although single-shot FPM techniques[24,25] have been proposed, these methods limit the field of view or spatial resolution and need an additional module, such as a lens array, which makes the optical setup unwieldy.

Coherent diffractive imaging (CDI) is a single-shot non-interferometric method that is extensively utilized in X-ray imaging[26–28]. CDI measures the intensity in the Fourier plane (or the far-field diffraction plane) while limiting the optical field to geometric support in the sample plane[29]. The Fourier spectrum has strong power at low frequencies. It may miss low spatial frequency information in the vicinity of zero spatial frequency[30–32]. In addition, imposing a sample support

---

[1]Department of Physics, Korea Advanced Institute of Science and Technology (KAIST), Daejeon 34141, Republic of Korea. [2]KAIST Institute for Health Science and Technology, Daejeon 34141, Republic of Korea. [3]Tomocube, Inc., Daejeon 34051, Republic of Korea. ✉e-mail: yk.park@kaist.ac.kr

limits the types of samples that can be imaged and requires prior knowledge of the sample[33,34]. The deviation in the sample spectrum from the support can cause significant errors in the reconstructed field. Recently, efforts have been made to alleviate the strict support condition of CDI[35,36]. For example, a loosened support condition was presented for imaging of separated objects[35]; however, this method can still target only certain types of samples corresponding to separated objects.

This study circumvents these drawbacks of CDI by limiting the field to a known support in the Fourier plane. The principle and optical setup are reciprocal to those of CDI. Hence, we termed it as "reciprocal diffractive imaging (RDI)." In RDI, a single mask placed in the Fourier plane filters the sample beam, and the intensity distribution is measured in the image plane. For diffusive samples, the Fourier spectrum satisfies the tight support edge condition of Fienup's hybrid input–output (HIO) algorithm[29] even for an elementary Fourier mask. This configuration enables a significantly simpler realization than similar modalities[37]. RDI neither restricts the sample plane nor requires a high dynamic range or sensitivity for a detector. This is because the imaging is conducted in image space. The Fourier mask is not intrusive. Here, we demonstrate RDI for various 3D objects with optically rough surfaces in the reflection geometry.

## Results
### Principle
RDI switches the constraints from the image plane to the Fourier plane. Figure 1 shows the optical setup used to image a diffusive object using RDI. The diffusive object is first illuminated with coherent light. In the diffusive regime, for light reflected from a diffusive object to cause interference, the incident light must possess temporal and spatial coherence. This is due to the lack of correlation between the wavefronts of incident light and diffused light. Then, the scattered light is filtered in the Fourier plane before being relayed to a camera by a 4-$f$ telescopic imaging system. A specially designed mask is placed in the Fourier plane and imposes a support constraint.

The Fourier mask defines the resolution of an optical system by blocking the light outside the support. When the light intensity is recorded, according to the Nyquist–Shannon sampling theorem, the length of a side of the mask $D$ should satisfy $f_2 \cdot w/D \geq 2p$, where $f_2$, $w$, and $p$ are the focal length of the second lens, the wavelength of the light source, and the camera pixel pitch, respectively. Considering this condition, we determined the size of the mask. Earlier research has indicated that the mask should have a non-centrosymmetric shape for the convergence of the RDI algorithm[38–40]. If the mask is perfectly square, which is centrosymmetric, Fienup's HIO algorithm may not provide a correct solution for a complex field[30]. Hence, we adopted a square mask with a slightly cropped corner to minimize intensity and resolution losses while providing isotropic imaging characteristics in every direction. Note that our method's applicability is not limited to this particular shape of the used Fourier mask. The Fourier mask, if asymmetric, can be freely designed[41] to best fit the experimental requirements. For instance, if a particular direction exhibits unwanted specular reflection, it can negatively impact the reconstruction process. However, this issue can be resolved by adjusting the shape of the Fourier mask to block the reflection in the Fourier plane. Also, in the presence of large noise, the size of the cropped corner may need to be increased for robust reconstruction under experimental noise[42,43]. A support of a triangular shape is significantly robust to noise[33].

Finally, the camera captures the intensity image of the scattered light from which the optical field is reconstructed. Details for the experimental setup are described in the "Methods" section.

The proposed method employs the reciprocal version of Fienup's hybrid input-output algorithm. The sample field to be reconstructed is $f(x, y)$. $F(u, v)$ denotes its two-dimensional (2D) Fourier transform. We set a support

$$s(u,v) = \begin{cases} 1, & (u,v) \in S \\ 0, & (u,v) \notin S \end{cases} \tag{1}$$

in Fourier space. Here, $S$ is the set of transmitted frequencies corresponding to the Fourier mask. The part where light passes through the Fourier mask in Fig. 1 represents the part where $s(u, v) = 1$; on the other

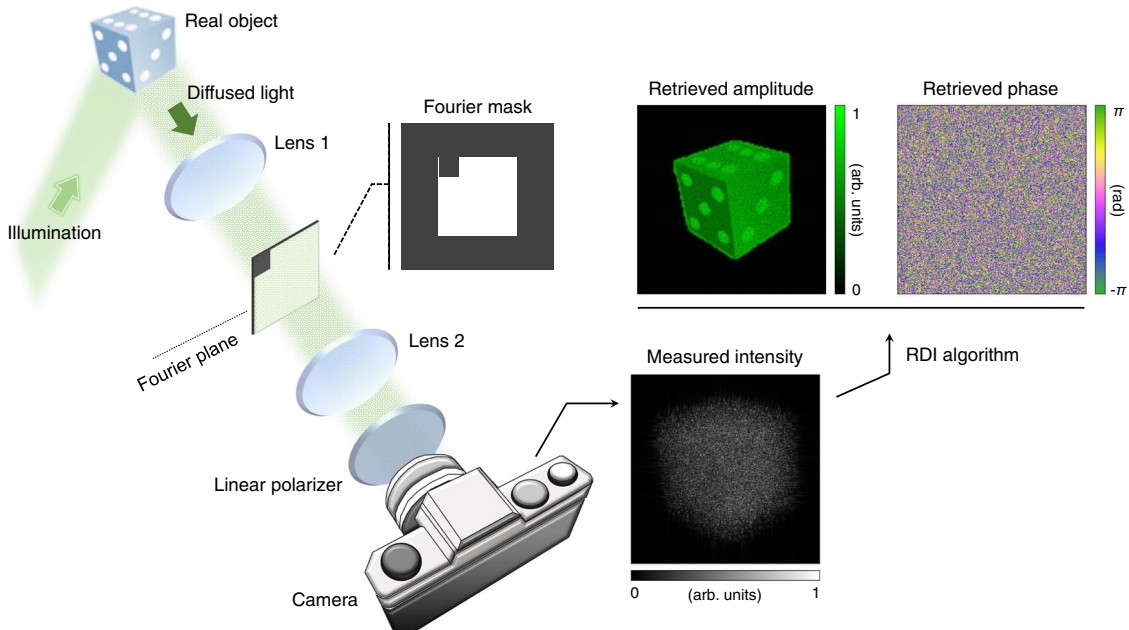

**Fig. 1 | Schematics of reciprocal diffractive imaging (RDI).** A plane wave illuminates an object. The diffused light passes through 4-$f$ relay lenses and a Fourier mask. The modulated light is polarized by a linear polarizer and impinges on a camera. The camera measures the image intensity. The optical field is retrieved from the intensity profile using the RDI algorithm.

hand, the part where light is blocked corresponds to $s(u, v) = 0$. After filtering in Fourier space, the optical field can be described as

$$F'(u,v) = F(u,v)s(u,v) \qquad (2)$$

The magnitude of the inverse transform of the modulated Fourier field $F'(u,v)$ is measured in a detector. This provides a constraint in real space

$$|f'(x,y)| = |\mathcal{F}^{-1}[F'(u,v)]| \qquad (3)$$

where $\mathcal{F}^{-1}$ denotes the 2D inverse Fourier transform. Using the constraints of $s(u, v)$ and $|f'(x,y)|$ in both Fourier and real spaces, the sample field is reconstructed iteratively. An input field in Fourier space at the $k$-th iteration step, $G_k(u, v)$, is transformed to $g_k(x, y)$ using the inverse Fourier transform. The field in real space with the constraint $|f'(x,y)|$ is defined as

$$g'_k(x,y) = |g'_k(x,y)| \exp(i\arg(g_k(x,y))) \qquad (4)$$

where $|g'_k(x,y)|$ is substituted by $|f'(x,y)|$ with a small variation to suppress the noise (see the Methods section). The Fourier transform of $g'_k(x,y)$, denoted by $G'_k(u,v)$, determines the output field of the iteration with the support constraint in Fourier space as follows:

$$G_{k+1}(u,v) = \begin{cases} G'_k(u,v), & (u,v) \in S \\ G_k(u,v) - \beta\, G'_k(u,v), & (u,v) \notin S \end{cases} \qquad (5)$$

where $\beta$ is the feedback parameter. It implies that a support constraint is not rigidly imposed. The condition for the background of the support, where $s(u, v) = 0$ occurs, is gradually reflected to prevent discontinuity between iterations. The field $G_{k+1}(u,v)$ becomes the input field for the next iteration. In this work, the initial guess of iteration was chosen from a random field whose amplitude was equal to that of the measured modulus. The value of $\beta$ was set to 0.6[29]. Finally, the iterations generate the convergent solution of $F'(u,v)$. The 2D inverse Fourier transform of $F'(u,v)$ is the low pass filtered form of $f(x, y)$.

## Experimental validation for diffusive objects

We verified the capacity of the RDI camera by imaging diffusive objects: the letters "EN," a university logo, a printed United States Air Force (USAF) resolution target, and a one-dime coin, as shown in Fig. 2. Focused photographs of the diffusive objects are also presented. The intensity images of the defocused scattered field blur and were difficult to recognize because the field propagated far from the focal plane. The amplitude and phase of the field retrieved from the measured intensity images were refocused (see the Methods section for numerical refocusing). Once refocused, the field showed the object details. This demonstrates the ability of RDI to recover both the amplitude and phase of an optical field. The refocused amplitude and intensity images exhibited the shape of a focused photograph. Nonperiodic or nonbinary samples were successfully reconstructed. A well-known feature of the USAF resolution target was resolved. Although the coin can be treated as a reflective sample, its pattern was distinctly visible in the retrieved amplitude and intensity. The phase of the fields appeared

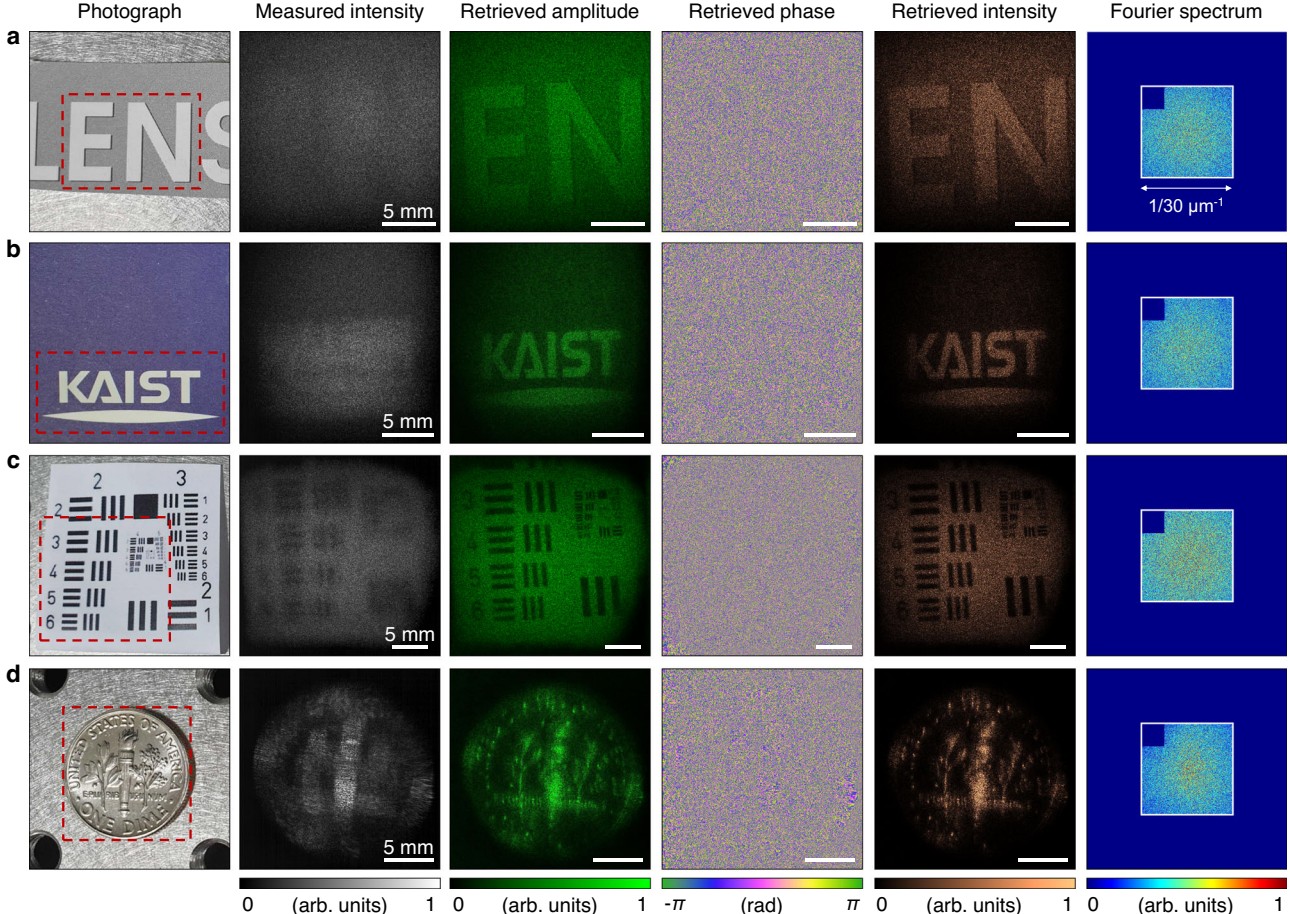

**Fig. 2 | Experimental validation. a–d** The light scattered from various objects was retrieved by the RDI algorithm: the letters "EN" (**a**), a university logo (**b**), a printed United States Air Force resolution target (**c**), and a one-dime coin (**d**). The photographs, measured intensities, retrieved amplitudes, retrieved phases, retrieved intensities, and reconstructed Fourier spectra are shown. The red dotted box in the photographs indicates the imaged area.

random because of the optical roughness of the samples. Nevertheless, the retrieved phase contained the necessary information to enable refocusing.

The maximum field of view (FOV) was $28.16 \times 28.16$ mm$^2$. The resolution of the optical system was 30 μm. It was determined by the size of the Fourier mask and the focal length of the lens. The resolution in the camera plane was negligibly more than twice the camera pixel pitch according to the Nyquist–Shannon sampling theorem. This is because the bandwidth of the intensity at the camera plane is twice the bandwidth of the field. In the reconstructed Fourier spectra of Fig. 2, the area of the support corresponding to the inverse of a resolution of 30 μm is marked, and it can be seen that the Fourier spectra are filled in this area except for the cropped corner. The Fourier spectra show the amplitudes of the Fourier field with a linear colormap. Reconstructing with imaging conditions of 30 μm resolution and a $15 \times 15$ mm$^2$ field of view, using a graphics processing unit (GeForce RTX 4090, NVIDIA Corp.), the total reconstruction time remained within 16 s, and memory usage was less than 50 Mb. The low memory requirement is mainly attributed to the fact that the two Fourier transforms of a field account for the majority of the operations in each iteration. We anticipate that advanced parallel processing and recently improved CDI algorithms can significantly reduce the reconstruction time[44,45].

To further demonstrate the capability of the proposed holographic camera, diffusive objects located at various distances were imaged. The resultant scattered field was refocused using the numerical propagation method. Figure 3a shows a schematic of the optical system and the location of $10 \times 10 \times 10$ mm$^3$ dice that were illuminated by a plane wave. The intensity of the diffused light was measured using a camera (Fig. 3b, c). The two dice in FOV1 were placed at different distances such that the focus positions for each dice were different. Numerical refocusing results (retrieved amplitude, phase, and intensity) are shown at intervals of 50 mm: FOV1 ranged from −150 to +150 mm (Fig. 3d–f), and FOV2 ranged from +50 to +350 mm (Fig. 3g–i). The retrieved intensity image was represented along with the amplitude image because the intensity is the contrast our eye perceives. At −100 mm, one dice in FOV1 looked sharp, and the other in FOV1 was blurred. At +50 mm, the situation was reversed. The dice in FOV2 were difficult to recognize in the vicinity of 0 mm and were sharply focused between +200 and +250 mm. These observations suggest that the holographic camera can freely adjust the focus, even for an out-of-focus object, and refocusing provides new information not available at other depths. The results for the numerical refocusing of the dice from −300 to +300 mm can be seen in Supplementary Video 1.

## Real-time imaging of moving objects

Moving objects were imaged to demonstrate the advantages of single-shot image acquisition. The fast motion of diffusive objects can be monitored using RDI. FPM[23] requires multiple acquisitions and is incompatible with real-time imaging. In our experiment, image acquisition was conducted when the letters "AB" were translated (Fig. 4a). The letters appeared and disappeared consecutively in the field of view (Fig. 4c). Next, imaging was conducted while rotating $10 \times 10 \times 10$ mm$^3$ dice (Fig. 4b). As the dice rotated, the different faces of the dice were captured over time (Fig. 4d). From the single-shot nature of RDI, we could reconstruct the optical fields without blurring the image owing to motion. The letters "AB" and dice were imaged for 16.47 and 22.35 s at a frame rate of 4.25 fps, respectively (see Supplementary Videos 2 and 3). This frame rate was the maximum at the field of view used and the bit depth of the camera. It can be controlled depending on the experimental situation. While the objects were moved by hand, the speed was estimated to be approximately 1 mm/s for the letters "AB" and dice. Although the speed of the objects was not constant, the exposure time used in the experiments was 3.73 ms. As an object may move slightly during the exposure time, shorter exposure times are generally preferred. If the light source has sufficient intensity, the exposure time can be reduced further, depending on the camera specification. Note that the frame rate was primarily restricted by the bandwidth of the camera-computer link, rather than the exposure time. The refocused distance is 100 mm in Fig. 4c, d.

Figure 5 shows the applicability of the proposed method. Several scenes in which dolls posed were filmed and made into an animation. The experimental scheme is depicted in Fig. 5a. Whenever the scene containing the actors' movement changed, a camera took the image. A total of 193 scenes were captured within 2 min. The actors were moved at varying speeds, ranging approximately from 1 to 10 mm/s. In the case of a conventional camera (the defocused distance was 150 mm), when an image of defocused objects was taken, a sharp image could not be obtained (Fig. 5b). On the contrary, the proposed holographic camera could get focused (Fig. 5c) and defocused (Fig. 5d, e) images of the actors using refocusing. The actors in each scene were exuberantly exhibited. This application forecasts that the proposed method will be widely used in the film industry and visual media as well. The produced animation can be seen in Supplementary Video 4.

## Suppression of speckle noise using angular compounding

Speckle noise caused by the diffusive nature of objects can be reduced using the angular compounding method. This method indicates spatial averaging of the retrieved amplitudes while changing the incident angle of illumination. We implemented angular compounding by locating a diffuser behind plane wave illumination and acquiring the intensity images while the diffuser was rotated (Fig. 6a). A granular speckle pattern was observed in the amplitude of the field retrieved from a single-shot intensity image (Fig. 6b). On the other hand, Fig. 6c shows the result of averaging 15 reconstructed amplitude images (Fig. 6c). It could be seen that speckle noise was greatly suppressed, and the image quality was improved. Fine features that are not discerned in Fig. 6b appear in the angular compounding results. The refocused images of the results are also shown. To quantitatively evaluate the reduction of speckle noise according to the averaging number of the retrieved amplitudes, the standard deviation of the amplitude was calculated in the red-line box in Fig. 6. The theoretical standard deviation of speckle noise was calculated by taking the experimental standard deviation obtained without averaging as the initial value and multiplying it by the inverse of the square root of the averaging number. Figure 6d shows the agreement between the prediction and the results of the quantitative analysis.

## Discussion

In summary, we reconstructed the field scattered from diffusive objects by exploiting the reciprocity from coherent diffractive imaging. We demonstrated the versatility of the proposed method by conducting imaging on various objects. By obtaining the amplitude and phase of the optical field, the image of the object can be freely refocused. Field retrieval is possible with only one intensity measurement. Hence, the dynamics of fast-moving objects can be observed without motion blurring. The holographic camera can be attached to the front of a generic camera and used for various electromagnetic spectra, such as IR, UV, and X rays. For example, the present method can be expanded in the X-ray domain to extract quantitative phase delay images of low-contrast samples by incorporating a Fourier plane into an existing X-ray imaging setup. The potential of this method can be fully realized through practical implementations for real-world applications, one of which we give here; RDI enables the reconstruction of 3D profiles of remote objects, even in challenging environments with atmospheric turbulence, vibrations, and mechanical instabilities, improving remote vision and light detection and ranging (LiDAR) and benefiting autonomous vehicles, robots, and environmental monitoring[46,47].

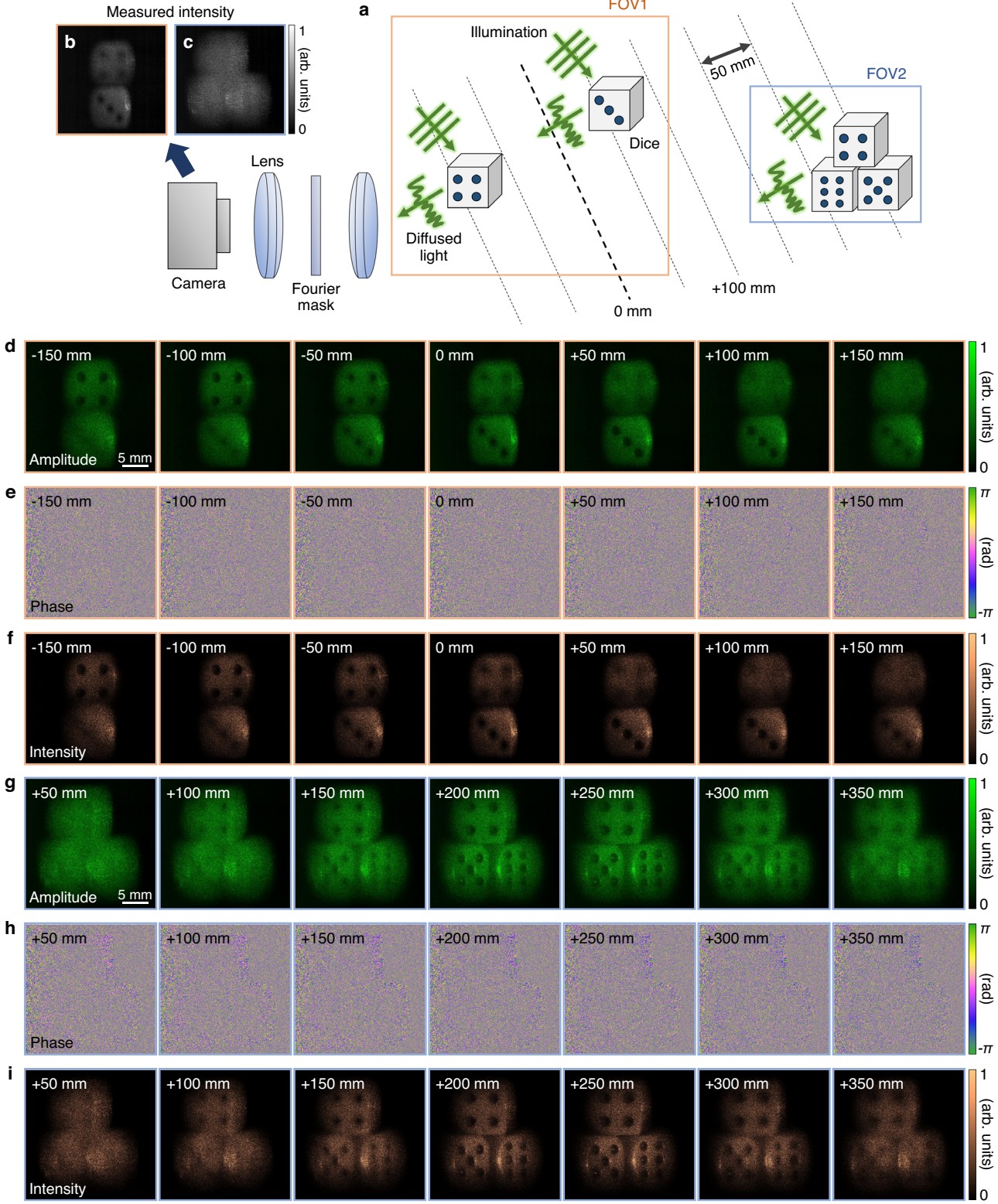

**Fig. 3 | Numerical refocusing of the diffusive field from dice. a** Experimental schematic. Dice are placed at different distances. Coherent light is incident on the dice, and the scattered light is measured by a camera. **b, c** Acquired intensity images for the field of views FOV1 (**b**) and FOV2 (**c**). **d–i** The retrieved field of the dice was numerically propagated, and the amplitude, phase, and intensity are shown for FOV1 (**d–f**) and FOV2 (**g–i**). FOV1 and FOV2 are represented by orange and blue colors, respectively. The refocused distance is marked as inset.

It should be emphasized that the principle and capability of the proposed method are distinguished from a light field camera[48–50]. The light field camera provides the refocusing capacity from a captured image. However, the technique uses a finite number of microlenses; therefore, it is impossible to recover all the fields for continuous coordinates[49]. This caveat significantly limits the spatial resolution and axial depth of the imaging system.

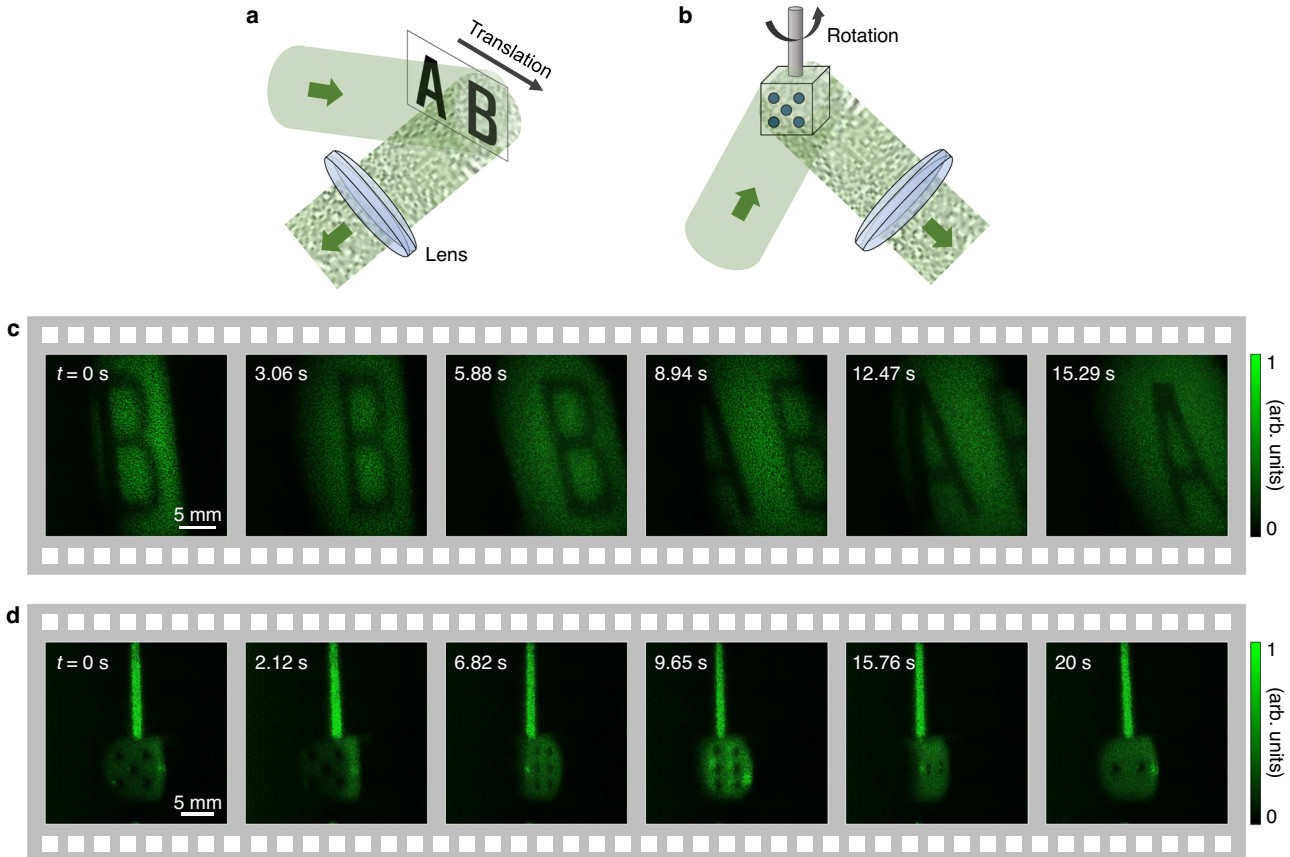

**Fig. 4 | Field retrieval of moving objects. a, b** Coherent light is incident on moving objects: translating the letters "AB" (**a**) and rotating dice (**b**). **c, d** Snapshots (field amplitudes) were taken over time for the letters "AB" (**c**) and dice (**d**).

Various non-interferometric holographic methods, such as conventional CDI[26,51,52], differential phase contrast microscopy[53,54], transport intensity equation methods[55,56], and Kramers–Kronig holographic imaging[57,58], have been developed. However, the studies have made assumptions regarding the imaged sample or the illumination. In particular, the weak scattering assumption hinders the application of many quantitative phase imaging methods to holographic cameras for diffusive samples. For example, optical diffraction tomography or 3D quantitative phase imaging[8,17,59] is remarkable in 3D imaging of transparent samples in a transmission-type manner but is not adequate for the reconstruction of the light reflected from a diffusive object. By contrast, the proposed holographic method excels in imaging speckle fields using a simple configuration. We envision that the stand-alone holographic camera could be utilized to vividly capture real life.

## Methods

### Experimental setup

The experimental setup is illustrated in Fig. 1. The light source is a spatially filtered coherent laser (531.65 nm, Laserglow Technologies, S533001FX). The light scattered from a sample passes through a 500 mm lens, Fourier mask, and 200 mm lens. The Fourier mask is custom-made. The part through which the light passes is $8.867 \times 8.867$ mm$^2$. The cropped corner is $2.217 \times 2.217$ mm$^2$. The light intensity is measured using a camera (pixel size = 5.5 μm, XIMEA, MQ042MG-CM). A linear polarizer (Thorlabs, Inc., LPVISE100-A) is placed in front of the camera. In the angular compounding experiments, a holographic diffuser (Edmund Optics Inc., 54-493) is used. The dolls presented in Figs. 5 and 6 were acquired from Preiser USA.

### Noise-robust RDI algorithm

With noise, the moduli in real space can cause the value of the background pixels (outside the support region) in Fourier space to be nonzero, even if the correct phase values are estimated. This discordance between the constraints of the two spaces causes oscillations in the iterative steps of the HIO algorithm and hinders the convergence to the correct solution[60,61]. To alleviate the convergence issue caused by noise, the noise-robust HIO algorithm was used in this study based on the method presented in Ref. 61. In Eq. (4), the modulus of $g'_k(x,y)$ is replaced by

$$|g'_k(x,y)| = (1 - \lambda)|f'(x,y)| + \lambda|g_k(x,y)| \qquad (6)$$

rather than $|g'_k(x,y)| = |f'(x,y)|$, where $\lambda$ is the relaxation parameter. This noise reduction algorithm renders $g'_k(x,y)$ resemble $g_k(x,y)$ more than $f'(x,y)$. The constraint imposed by $|f'(x,y)|$ is less rigidly reflected in each iteration step. Therefore, this method is known as the "constraint relaxation method."

Two different values of $\lambda$ were chosen according to the number of iterations. When the number of iterations is small (≤500), $\lambda$ prevents oscillations originating from the violation of the constraints of Fourier space and real space. For a large number of iterations (>500), $\lambda$ filters the noise in the modulus image. The value of $\lambda$ in this study was set to 0.005 until the number of iterations was 500 and to 0.1 thereafter[61].

### Numerical refocusing process

Numerical refocusing, also known as the angular spectrum propagation method, is conducted by applying the "propagation kernel" to the Fourier spectrum in Fourier space in order to propagate a complex field[62,63]. If a complex field $f$ propagates as the distance $z$ in free space,

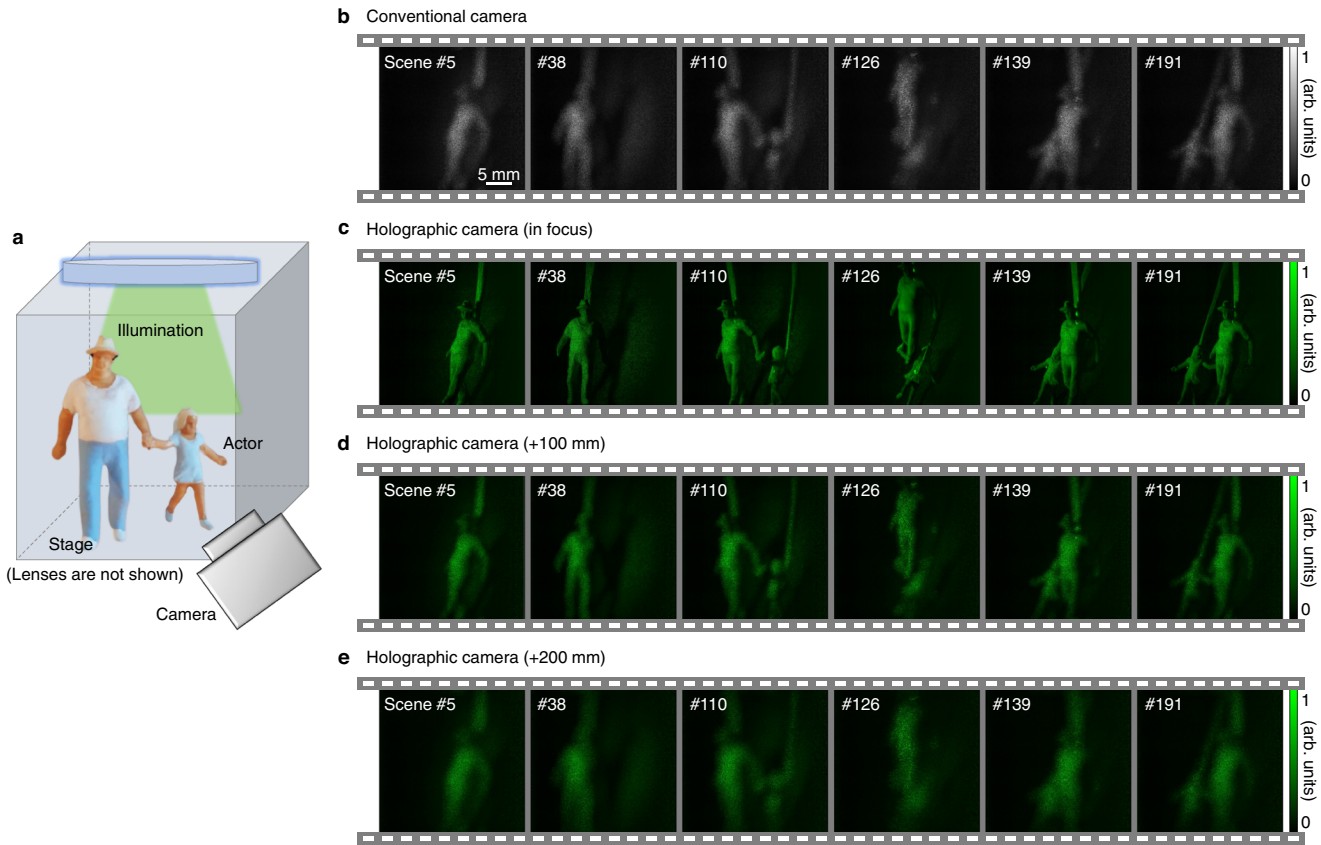

**Fig. 5 | Frames of animation with dolls. a** An illumination beam is incident on dolls (actors), and the scattered beam is collected by a camera. The actors are in various poses. **b** Scenes with the actors were filmed via a conventional camera. **c**–**e** The proposed holographic camera imaged the field amplitude for the same scenes with different refocusing distances: in focus (**c**), 100 mm (**d**), and 200 mm (**e**).

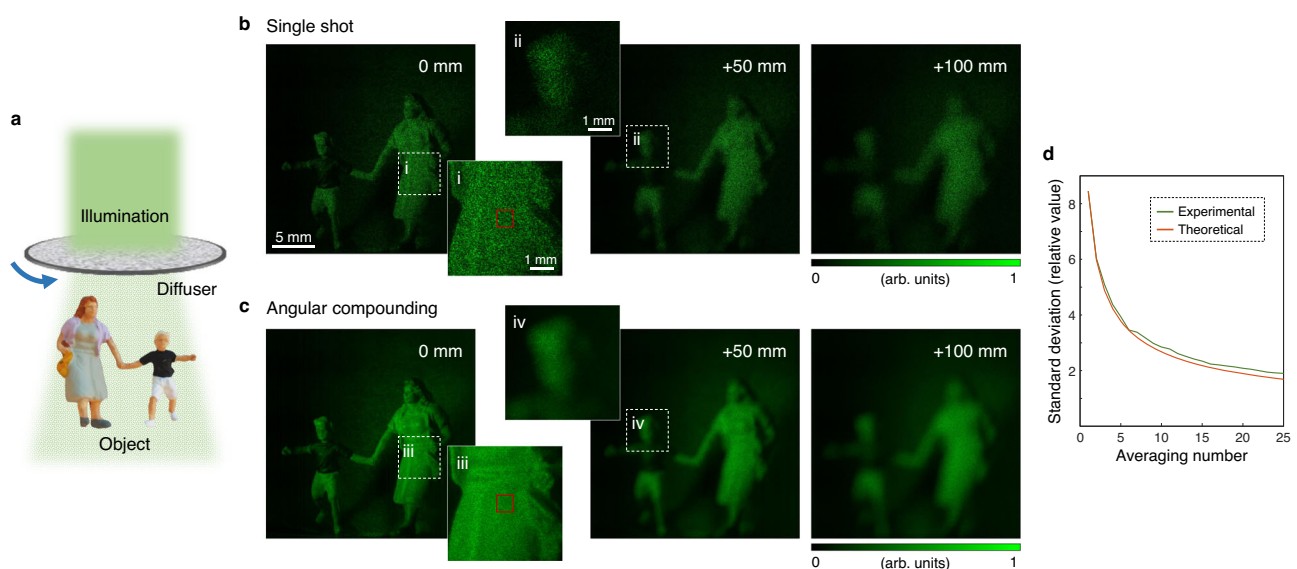

**Fig. 6 | Comparison of single-shot and angular compounding results. a** A diffuser is located after a plane wave, making speckle illumination impinge on an object. For each measurement, the diffuser is rotated. **b** Field amplitude for each refocused distance. **c** Angularly compounded results for each refocused distance. The dotted-line boxes "i–iv" are enlarged to inset. The refocused distance is marked as inset. **d** Plots of the experimental and theoretical standard deviations according to the averaging number in the spatial domain indicated by the red-line box.

the 2D Fourier transform of $f$ is represented as:

$$\mathcal{F}[f(x,y,z=d)](u,v,z=d) = \mathcal{F}[f(x,y,z=0)]e^{2\pi i d w}$$
$$= \mathcal{F}[f(x,y,z=0)]e^{2\pi i d\sqrt{(1/\lambda)^2 - u^2 - v^2}} \qquad (7)$$

where $\mathcal{F}$ denotes the 2D Fourier transform, $(u, v, w)$ are the spatial frequency coordinates corresponding to $(x, y, z)$, and $e^{2\pi i d\sqrt{(1/\lambda)^2 - u^2 - v^2}}$ is called the propagation kernel. The propagated field in image space is then derived by applying the 2D inverse Fourier transform to Eq. (7).

## Data availability
The data generated in this study have been deposited in our GitHub repository: github.com/BMOLKAIST/Reciprocal-Diffractive-Imaging-2023.

## Code availability
The MATLAB code for the implementation of RDI is available at github.com/BMOLKAIST/Reciprocal-Diffractive-Imaging-2023.

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

## Acknowledgements

This work (J.O., H.H., and Y.P.) was supported by KAIST UP Program, BK21+ Program, Tomocube, National Research Foundation of Korea (2015R1A3A2066550, 2022M3H4A1A02074314), and Institute of Information & communications Technology Planning & Evaluation (IITP; 2021-0-00745) grant funded by the Korea government (MSIT).

## Author contributions

J.O. and Y.P. conceived the project. J.O. and H.H. contributed to the development of the mathematical framework and the analysis tools. J.O. conducted the experiment and the analysis. All authors wrote the manuscript. Y.P. provided supervision.

## Competing interests

The authors declare no competing interests.
