## [Peer Review File · Nature Communications]

REVIEWER COMMENTS

Reviewer #1 (Remarks to the Author):

The authors report about a method for single-shot holographic imaging, without a separate interferometric reference wave for hologram recording, in which the object wave is reconstructed iteratively from laser light illuminated specimens that are imaged via a non-symmetric phase mask within the Fourier plane of a 4-f telescopic imaging system onto digital image recording device. After an introduction to the underlying principles, the performance of the proposed method is illustrated by experimental results from test targets, dice, and dolls.

In general, the work is motivated organized and includes adequate references. The experimental investigations appear to be accurately performed. The presented results are plausible and novel. The authors address an important topic in digital holography: The single-shot recording, reconstruction, and refocusing of scattered wave fields / images of investigated specimens in a simplified setup without the need of an interferometric reference wave, which thus is insensitive of mechanical vibrations. The content of the manuscript may thus be of high interest for the field of digital holography and related areas such as non-destructive testing.

However, the authors may consider revisions:

1. *Title*: An essential aspect of the proposed method seems to be that no interferometric reference wave is utilized in the experimental setup. For clarity, the authors may consider this topic in the title.
2. *Abstract*: Line 15: Instead of “complicated”, “complex” appears to be more appropriate.
3. *Introduction*: The authors should consider describing their work in relation with further work that appears to be closely related to the content of the manuscript, e.g., Nat. Commun. 7, 10820 (2016), Optica 5, 976-983 (2018), and should briefly discuss differences/similarities and advantages/limitations to the proposed single-shot concept in context of this earlier reported work.
4. *Principle / Experimental setup*:
 - a. Line 73: The “specially designed mask” and its influence in the recording and the numerical reconstruction process should be explained and discussed with more details.
 - b. Lines 79-80: The mean of “... to best fit experimental requirements” should be explained and discussed with more details.
5. *Theoretical description of the recording / reconstruction process*: From the explanations of Eq. 1 it becomes not fully clear if the parameter s is identical with the “Fourier mask” in Fig. 1. Additional clarifying information should be added by the authors.
6. *Numerical evaluation of the recorded intensity patterns*:
 - a. Line 115: From the descriptions it becomes not fully clear how numerical refocusing was performed. The authors should add additional clarifying information and an adequate reference for the utilized algorithms.
 - b. The computation times / computer capacity demands for the image reconstruction process should be specified and briefly discussed.
 - c. The required coherence properties of the applied laser / laser light should be briefly discussed.
7. *Experimental results*:
 - a. In line 125 the authors write that the optical resolution amounts to 30 μm but it becomes not fully clear if the resolution was also achieved also experimentally, e.g., by the USAF test chart in Fig. 2c. The authors should add information/explanations to clarify this topic.
 - b. Section “Real-time imaging for moving objects”: The author may change “for” to “of” in the section title. Moreover, in the descriptions of the section, the authors should add additional information about the experimental parameters, e.g., concerning the speed of the observed moving objects and the exposure time that was utilized/achievable during the recording of the intensity patterns.
 - c. Fig. 6d: Additional explaining information about the “theoretical standard deviation” should be provided.
8. *Discussion and outlook*:
 - a. The authors should add a more detailed discussion concerning the prospective usage of the

retrieved phase information for metrology applications and the possibly expected possible benefits o of their approach which could significantly improve the impact of the manuscript.

- b. In line 192 the authors write that their imaging concept may be also applicable for X-rays which seems to require a significantly different imaging/detectpr system. The authors should add further explaining information to clarify this topic.

Reviewer #2 (Remarks to the Author):

The paper is well written, methodology is well described, and the results are convincing. I think it will advance the field, so I would recommend publishing it. The only suggestion (which may be just a personal preference) is to perhaps expand the introduction and discuss how this method and its results compare to propagation-based diffraction tomography. There are several papers (e.g., references 16, 17), which can be discussed a bit more.

Point-by-point response to the reviewers' comments

We thank the reviewers for their constructive comments, which were very helpful in improving our manuscript. We have addressed all the reviewers' points, and the revised sections are summarized in the table below. The original referee comments are provided in black color, whereas our responses are given in blue. The appropriate changes made in the revised manuscript are highlighted. We have also taken the opportunity to make some other changes in order to refine the quality of the manuscript; these are also highlighted.

Sections revised	
Title	● Addition of the term Noninterferometric to the title
Abstract	● Modification to a more appropriate word
Introduction	● Discussion for related works● Discussion for propagation-based diffraction tomography
Results	● Elaboration on the role of the designed mask in the recording and the numerical reconstruction● Explanation for the phrase “to best fit experimental requirements”● Clarification of the parameter s regarding the Fourier mask● Elaboration on the process of the numerical refocusing● Addition of information on the computational time and capacity in the reconstruction● Addition of information on the required coherence properties of a light source● Description for the spatial resolution of the optical system● Revision of Fig. 2● Modification of the subtitle● Addition of information on the experimental parameters● Clarification of the term theoretical standard deviation
Discussion	● More detailed discussion for prospective usage and possible benefits of the proposed method● Explanation for the employment of the proposed method in other electromagnetic spectra
Supplement	● Dissemination of the code and data

Reviewer #1 (Remarks to the Author):

The authors report about a method for single-shot holographic imaging, without a separate interferometric reference wave for hologram recording, in which the object wave is reconstructed iteratively from laser light illuminated specimens that are imaged via a non-symmetric phase mask within the Fourier plane of a 4-f telescopic imaging system onto digital image recording device. After an introduction to the underlying principles, the performance of the proposed method is illustrated by experimental results from test targets, dice, and dolls.

In general, the work is motivated organized and includes adequate references. The experimental investigations appear to be accurately performed. The presented results are plausible and novel. The authors address an important topic in digital holography: The single-shot recording, reconstruction, and refocusing of scattered wave fields / images of investigated specimens in a simplified setup without the need of an interferometric reference wave, which thus is insensitive of mechanical vibrations. The content of the manuscript may thus be of high interest for the field of digital holography and related areas such as non-destructive testing.

We sincerely appreciate the reviewer for the careful evaluation and consideration of our work.

However, the authors may consider revisions:

1. Title: An essential aspect of the proposed method seems to be that no interferometric reference wave is utilized in the experimental setup. For clarity, the authors may consider this topic in the title.

Thank you for the suggestion. We have changed the previous title to “*Non-interferometric stand-alone single-shot holographic camera using reciprocal diffractive imaging.*”

2. Abstract: Line 15: Instead of “complicated”, “complex” appears to be more appropriate.

Thank you for the comment. We have changed “complicated” to “complex” in the abstract.

3. Introduction: The authors should consider describing their work in relation with further work that appears to be closely related to the content of the manuscript, e.g., Nat. Commun. 7, 10820 (2016), Optica 5, 976-983 (2018), and should briefly discuss differences/similarities and advantages/limitations to the proposed singleshot concept in context of this earlier reported work.

We appreciate the reviewer for the recommendation of useful references. We would like to discuss the differences/similarities and advantages/limitations of the previous methods as follows.

(1) *Nature Communications* 7, 10820 (2016): This study exploits a similar idea to the paper “Proposal for phase recovery from a single intensity distribution (doi.org/10.1364/OL.1.000010).” The authors successfully reconstructed the complex field scattered from separated objects. However, the method still has the limitations of conventional coherent diffractive imaging (CDI). The acquisition of intensity is conducted in Fourier space, which makes the reconstruction vulnerable to the signal-to-noise ratio and saturates the intensity around the DC term. Also, separated objects can be regarded as a kind of a support. Although the imaging system requires a loosened support condition for each object, the method can still image only certain types of samples corresponding to separated objects.

(2) *Optica* **5**, 976-983 (2018): The authors conducted single-shot phase imaging using a lens array and the Fourier ptychographic algorithm. The method is promising and well demonstrated, but it has several disadvantages. Because the Fourier ptychographic method is used, overlapping in Fourier space (redundant data) is required, and the reconstruction is vulnerable to misalignment and only works for thin samples. Also, for the sake of imaging with a single frame, the field of view should be significantly decreased along with the requirement of redundant data. In the paper, the authors divided the camera plane into 21×21 tiles. Moreover, the method has the same pitfalls as imaging methods using a lens array: limited spatial resolution; complex configuration; requiring sophisticated manufacturing and alignment; significant aberration; crosstalk between contiguous lens elements.

We have added the recommended references and related discussion to the introduction.

4. Principle / Experimental setup: a. Line 73: The “specially designed mask” and its influence in the recording and the numerical reconstruction process should be explained and discussed with more details.

Thank you for your careful reading. The “specially designed mask,” called the Fourier mask in the manuscript, acts as a support in the Fourier plane. In terms of the recording process, the Fourier mask defines the resolution of the optical system by blocking the light outside the support. When the light intensity is recorded, according to the Nyquist–Shannon sampling theorem, the length of one side of the mask D should satisfy $f\lambda/D \geq 2p$, where f , λ , and p are the focal length of the second lens, the wavelength of the light source, and the camera pixel pitch, respectively. Considering this condition, we set the length of a side of the mask to 8.867 mm. The experimental parameters are described in the Method section in more detail.

In terms of the numerical reconstruction, the Fourier mask conducts a more important role. If the mask is perfectly square, which is centrosymmetric, Fienup’s hybrid input-output algorithm may not provide a correct solution for a complex field [Ref. 30]. Thus, in order to minimize the loss of information while making the mask asymmetric, the corner was cut off into the shape of a small square, as shown in Fig. 1. Any shape is possible as long as it is asymmetric, but as the asymmetry of the support increases, the reconstruction becomes more robust to experimental noise [Refs. 42, 43].

We have complemented the section.

b. Lines 79-80: The mean of “... to best fit experimental requirements” should be explained and discussed with more details.

We can give a few examples. If undesirable specular reflection appears strongly in a specific direction, it can adversely affect the reconstruction. This reflection can be blocked in the Fourier plane by fitting the shape of the Fourier mask. Also, in the presence of large noise, the size of the cropped corner may need to be increased for robust reconstruction under experimental noise. For example, a support of a triangular shape is significantly robust to noise [Refs. 42, 43]. We have complemented the explanation in the manuscript.

5. Theoretical description of the recording / reconstruction process: From the explanations of Eq. 1 it becomes not fully clear if the parameter s is identical with the “Fourier mask” in Fig. 1. Additional clarifying information should be added by the authors.

Thank you for the comment. The part where light passes through the Fourier mask in Fig. 1 represents the part where $s(u, v) = 1$ in Eq. (1); on the other hand, the part where light is blocked corresponds to $s(u, v) = 0$. We have clarified this point in the manuscript.

6. Numerical evaluation of the recorded intensity patterns: a. Line 115: From the descriptions it becomes not fully clear how numerical refocusing was performed. The authors should add additional clarifying information and an adequate reference for the utilized algorithms.

The numerical refocusing, also known as the angular spectrum propagation method, is conducted by applying the “propagation kernel” to the Fourier spectrum in Fourier space in order to propagate a complex field. If a complex field f propagates as the distance z in free space, the two-dimensional Fourier transform of f is represented as follows:

$$\mathcal{F}[f(x, y, z = d)](u, v, z = d) = \mathcal{F}[f(x, y, z = 0)]e^{2\pi i d w} = \mathcal{F}[f(x, y, z = 0)]e^{2\pi i d \sqrt{(1/\lambda)^2 - u^2 - v^2}},$$

where \mathcal{F} denotes the two-dimensional Fourier transform, (u, v, w) are the spatial frequency coordinates corresponding to (x, y, z) , and $e^{2\pi i d \sqrt{(1/\lambda)^2 - u^2 - v^2}}$ is called the propagation kernel. The propagated field in image space is then derived by applying the two-dimensional inverse Fourier transform to the above equation.

We have complemented this explanation and added appropriate references [Refs. 66, 67] in the manuscript.

b. The computation times / computer capacity demands for the image reconstruction process should be specified and briefly discussed.

Thank you for the comment. The total reconstruction time stayed within 16 s, and the required memory was less than 50 Mb for reconstruction with imaging conditions of 30 μm resolution and $15 \times 15 \text{ mm}^2$ field of view. A graphics processing unit (GeForce RTX 4090, NVIDIA Corp.) was used. The very low memory requirement is because the two Fourier transforms of a field account for most of the operations in each iteration. For the reconstruction time, we expect the time can be reduced greatly by employing advanced parallel processing and recently improved CDI algorithms [Refs. 44, 45]. We have added this information to the manuscript.

c. The required coherence properties of the applied laser / laser light should be briefly discussed.

The required coherence properties of our holographic camera can be considered divided into spatial and temporal coherences.

(1) Spatial coherence: It does not matter if the shape of the light incident on an object is a speckle (as presented in Fig. 6), but the incident light should be static within the exposure time of the detector, which means the light is needed to be spatially coherent. In fact, spatial coherence is required in most imaging methods that have conditions for the characteristic of the Fourier spectrum. This is because a detector measures the sum of each intensity of the optical fields generated within the exposure time, not the intensity of the sum of the fields. However, it is important that apart from the field retrieval method, the incident light should be spatially coherent, for the same reason discussed in the next paragraph.

(2) Temporal coherence: We measured a phase profile in the diffusive regime. The wavefront of the light reflected from a diffusive object is rapidly changed depending on the wavelength of the light source such as diffusive media. Thus, the incident light on a diffusive object should be temporally coherent to make the reflected light interfere. In general, in order for the light reflected from a diffusive object to cause interference, the incident light should be temporally and spatially coherent. This is because the wavefronts of incident light and diffused light do not generally correlate with each other. We have complemented this point in the manuscript.

7. Experimental results: a. In line 125 the authors write that the optical resolution amounts to $30\ \mu\text{m}$ but it becomes not fully clear if the resolution was also achieved also experimentally, e.g., by the USAF test chart in Fig. 2c. The authors should add information/explanations to clarify this topic.

We appreciate the reviewer for bringing attention to this point. The resolution was determined by the focal length of the first lens and the length of a side of the Fourier mask. The printed USAF target has been benchmarked for experimental verification of the proposed method. However, even if the resolution is actually $30\ \mu\text{m}$, it may be difficult to accurately confirm the corresponding features due to the quality of printing using printer ink, as shown in the photograph in Fig. 2c. Instead, we can demonstrate the resolution by showing the reconstructed Fourier space. We have added the images of the reconstructed Fourier spectra in Fig. 2 and the related discussion in the manuscript. The area marked in the Fourier spectra of Fig. 2 corresponds to the inverse of a resolution of $30\ \mu\text{m}$, and it can be seen that the Fourier spectra are filled in this area except for the cropped corner. The Fourier spectra show the amplitudes of the Fourier field with a linear colormap.

b. Section “Real-time imaging for moving objects”: The author may change “for” to “of” in the section title. Moreover, in the descriptions of the section, the authors should add additional information about the experimental parameters, e.g., concerning the speed of the observed moving objects and the exposure time that was utilized/achievable during the recording of the intensity patterns.

Thank you for your careful reading. We have corrected “for” to “of” in the subtitle.

In the case of moving objects, the speed is not constant because the objects were moved by hand, but the speed was approximately $1\ \text{mm/s}$ for the letters “AB” and dice in Fig. 4, and the dolls in Fig. 5 were moved with a speed range of approximately $1\text{--}10\ \text{mm/s}$. The exposure time utilized in the experiments was $3.73\ \text{ms}$, so the objects would have moved about $4\ \mu\text{m}$ during the exposure time. Because an object can move to some extent within the exposure time, the shorter the exposure time, the better, and if the intensity of the light source is strong enough, the exposure time can be reduced arbitrarily depending on the camera specifications. Note that the frame rate was mainly limited by the camera-computer link bandwidth rather than the exposure time.

We have provided this information in the manuscript.

c. Fig. 6d: Additional explaining information about the “theoretical standard deviation” should be provided.

The “theoretical standard deviation” according to the averaging number was calculated by taking the experimental standard deviation obtained without averaging as the initial value and multiplying it by

the inverse of the square root of the averaging number. We have clarified the term in the manuscript.

8. Discussion and outlook: a. The authors should add a more detailed discussion concerning the prospective usage of the retrieved phase information for metrology applications and the possibly expected possible benefits of their approach which could significantly improve the impact of the manuscript.

We agree that discussion for possible usage can enrich the contents of the manuscript. We have complemented more detailed discussions in the “Discussion” section.

b. In line 192 the authors write that their imaging concept may be also applicable for X-rays which seems to require a significantly different imaging/detector system. The authors should add further explaining information to clarify this topic.

Thank you for the comment. The present method can be expanded in the X-ray domain to extract quantitative phase delay images of low-contrast samples by incorporating a Fourier plane into an existing X-ray imaging setup. We have added this information to the manuscript.

We have strengthened the manuscript based on the reviewers’ comments and hope that the manuscript is now suitable for publication in *Nature Communications*. We again appreciate the reviewers for their valuable time and contribution to the manuscript.

Reviewer #2 (Remarks to the Author):

The paper is well written, methodology is well described, and the results are convincing. I think it will advance the field, so I would recommend publishing it. The only suggestion (which may be just a personal preference) is to perhaps expand the introduction and discuss how this method and its results compare to propagation-based diffraction tomography. There are several papers (e.g., references 16, 17), which can be discussed a bit more.

We sincerely appreciate the reviewer for positive feedback and evaluation of our work.

We think the discussion suggested by the reviewer is an important explanation. Propagation-based diffraction tomography is outstanding in imaging transparent samples in a transmission-type manner but is not adequate for the reconstruction of the light reflected from a diffusive object. This is because the principle of transmission-type diffraction tomography is not applicable in this regime, and diffraction tomography is generally based on a weak-scattering approximation. In addition, diffraction tomography usually requires an additional imaging module and dozens of intensity images, so it is far from the single-shot holography we claim. We have added this discussion to the “Discussion” section.

We have strengthened the manuscript based on the reviewers’ comments and hope that the manuscript is now suitable for publication in *Nature Communications*. We again appreciate the reviewers for their valuable time and contribution to the manuscript.

REVIEWERS' COMMENTS

Reviewer #1 (Remarks to the Author):

The authors have addressed my comments and questions.

Reviewer #2 (Remarks to the Author):

The authors have addressed all the comments and the manuscript is now suitable for publication in *Nature Communications*

We thank the reviewers for their constructive comments, which were very helpful in improving our manuscript. The original referee comments are provided in black color, whereas our responses are given in blue.

Reviewer #1 (Remarks to the Author):

The authors have addressed my comments and questions.

We sincerely appreciate the reviewer for the careful evaluation and consideration of our work.

Reviewer #2 (Remarks to the Author):

The authors have addressed all the comments and the manuscript is now suitable for publication in Nature Communications

We sincerely appreciate the reviewer for the careful evaluation and consideration of our work.